# Drug Resistance in Filarial Parasites Does Not Affect Mosquito Vectorial Capacity

**DOI:** 10.3390/pathogens10010002

**Published:** 2020-12-22

**Authors:** Erik Neff, Christopher C. Evans, Pablo D. Jimenez Castro, Ray M. Kaplan, Guha Dharmarajan

**Affiliations:** 1Savannah River Ecology Laboratory, University of Georgia, Drawer E, Aiken, SC 29802, USA; 2Department of Infectious Diseases, College of Veterinary Medicine, University of Georgia, Athens, GA 30602, USA; ccevans@uga.edu (C.C.E.); pablo.jimenezca25@uga.edu (P.D.J.C.); rkaplan@uga.edu (R.M.K.); 3Grupo de Parasitología Veterinaria, Universidad Nacional de Colombia, Bogotá 11001000, Colombia

**Keywords:** *Aedes* spp., arthropod, disease transmission, *Dirofilaria immitis*, drug resistance, extrinsic incubation period, filaria, macrocyclic lactones, mosquito, vector competence

## Abstract

Parasite drug resistance presents a major obstacle to controlling and eliminating vector-borne diseases affecting humans and animals. While vector-borne disease dynamics are affected by factors related to parasite, vertebrate host and vector, research on drug resistance in filarial parasites has primarily focused on the parasite and vertebrate host, rather than the mosquito. However, we expect that the physiological costs associated with drug resistance would reduce the fitness of drug-resistant vs. drug-susceptible parasites in the mosquito wherein parasites are not exposed to drugs. Here we test this hypothesis using four isolates of the dog heartworm (*Dirofilaria immitis*)—two drug susceptible and two drug resistant—and two vectors—the yellow fever mosquito (*Aedes aegypti*) and the Asian tiger mosquito (*Ae. albopictus*)—as our model system. Our data indicated that while vector species had a significant effect on vectorial capacity, there was no significant difference in the vectorial capacity of mosquitoes infected with drug-resistant vs. drug-susceptible parasites. Consequently, contrary to expectations, our data indicate that drug resistance in *D. immitis* does not appear to reduce the transmission efficiency of these parasites, and thus the spread of drug-resistant parasites in the vertebrate population is unlikely to be mitigated by reduced fitness in the mosquito vector.

## 1. Introduction

Vector-borne diseases remain an important challenge for public health globally, and contribute to over 17% of the global estimated burden of all infectious diseases [1]. Recent few decades have seen encouraging progress in the control of some important mosquito-borne diseases including malaria [2,3] and filariasis [4]. However, continued success in controlling these diseases faces numerous challenges, with the evolution of resistance against anti-parasitic drugs being one of the most important [5,6,7].

Given that the evolutionary dynamics of parasites are critically affected by the host environment (e.g., immune response [8]), vector-transmitted parasites are subjected to very different selection pressures in the different hosts (vertebrate and invertebrate) used to complete the life cycles. The differential pressures that parasites face in the vertebrate host vs. the arthropod vector could act to enhance or reduce parasite transmission, and thus impact the spread of drug-resistant parasite isolates. For example, the development of drug resistance in vector-transmitted parasites could have high fitness advantages for life history stages in the vertebrate hosts wherein they are likely to be exposed to chemoprophylaxis. However, drug resistance usually trades off against other fitness parameters (e.g., the ability of parasites to reproduce or evade the immune system), and thus drug-resistant genotypes are expected to have lower fitness in drug-free environments [9,10]. Thus, we expect that the drug-resistant strains may be less fit than their drug-susceptible counterparts in the vector, since parasites are not exposed to drugs in the mosquito.

If drug resistance trades off against physiological characteristics required by the parasite to infect the mosquito host, passage through the vector will select against drug-resistant parasite isolates. Indeed, previous studies in malaria parasites have shown that isolates resistant to the anti-malarial compound atovaquone have increased fitness in humans treated with the drug, but fail to develop, and hence be transmitted by the mosquito vector [11]. While drug resistance is an important factor affecting our ability to control filarial parasites globally, to our knowledge no studies have quantified the effects of drug resistance on the ability of a mosquito to transmit the parasite (i.e., vectorial capacity). In contrast to malaria parasites that generally have low fitness consequences in the natural vectors [12], filarial parasites generally induce high levels of mortality in mosquitoes [13,14]. Since vectorial capacity is a product of the number of parasites developing to the infectious stage and the probability of the vector surviving to transmit these stages to the vertebrate host [14], we expect that there will be complex interactions that ultimately determine whether drug resistance increases or decreases vectorial capacity. Consequently, to visualize these expectations, we built a model to simulate the effects of drug resistance and susceptibility on vectorial capacity (see Section 4 for details).

Our simulation model revealed that in the case of filarial parasites, the vectorial capacity of mosquitoes infected with drug-susceptible vs. drug-resistant parasite isolates would be affected by a complex interaction between three parameters: (a) establishment load: the initial load of establishing parasites; (b) vector efficiency: the proportion of establishing parasites that develop to L3; (c) survival to the extrinsic incubation period (EIP), which is the minimum period of time required for parasites to develop to their infectious stage in the vector (see Figure 1).

Specifically, in cases where drug resistance is associated with a large decrease in establishment load (Figure 1A,D,G), there would a significant reduction in vectorial capacity compared to drug-susceptible isolates, except in the case when drug resistance is also associated with a very large increase in vector efficiency (Figure 1A). Alternatively, when drug resistance is associated with a large increase in vector efficiency (but only a small reduction in establishment load), we expect higher vectorial capacity with drug-resistant vs. drug-susceptible isolates (Figure 1B). However, when a large increase in vector efficiency is combined with no reduction in establishment load, the high parasite load reduces survival to EIP, and thus is expected to reduce the vectorial capacity of drug-resistant vs. drug-susceptible isolates (Figure 1C). Small reductions in establishment load and/or increases in vector efficiency will have no detectable effects on vector capacity (Figure 1E,F,H). Consequently, contrary to the case of malaria, vectorial capacity is lower in drug-resistant vs. drug-susceptible parasites only under conditions of very low establishment load (Figure 1D,G) or very high vector efficiency (Figure 1C). This latter effect is due to the high fitness costs associated with filarial infection. Alternatively, under most other conditions, drug-resistant strains face few mosquito-related transmission costs (Figure 1A,E,F,H), and could even show higher transmission under some conditions compared to drug-susceptible strains (Figure 1B).

In this study, we focused on elucidating the effects of drug resistance on the vectorial capacity of mosquitoes infected with dog heartworm (*Dirofilaria immitis*). *Dirofilaria immitis* is a mosquito-transmitted filarial parasite that primarily infects dogs and other canids ([15,16]; see Section 4 or detailed life history characteristics). We used four isolates of the dog heartworm that were either susceptible or resistant to macrocyclic lactone anthelmintics (the primary chemotherapeutic approach to prevent *D. immitis* infections in dogs) as our model system. We studied parasite infection dynamics in two species of mosquitoes—the yellow fever mosquito (*Ae. aegypti*) and the Asian tiger mosquito (*Ae. albopictus*)—that are natural vectors of *D. immitis* [17,18] (see Section 4 for details). Drug resistance against macrocyclic lactones is a complex issue that compromises our ability to control *D. immitis* infections in dogs globally, and is an issue that is growing concern in the United States [19,20,21,22]. While numerous studies have documented the rise in resistance against macrocyclic lactones, the underlying genetic mechanisms driving this trait in *D. immitis* populations remains unclear [23,24]. Additionally, we also have little information on the potential fitness consequences associated with the drug-resistant phenotypes especially as it relates to their ability to be transmitted by the mosquito vector. A better understanding of the role that mosquitoes play in facilitating or inhibiting the spread of drug resistance *D. immitis* isolates during its life cycle has broad implications for developing transmission models and for implementing control strategies for filarial infections globally.

## 2. Results

### 2.1. Establishment Load

We found that despite standardization of mf concentrations to 5000 mf/mL in the infected blood used to feed mosquitoes, the initial establishment load differed by parasite isolate and vector isolate, with a significant interaction between these variables (Figure 2A; N = 118; parasite isolate: χ^2^ = 19.772; *p* < 0.001; vector isolate: χ^2^ = 8.694; *p* = 0.003; parasite isolate × vector isolate: χ^2^ = 3.716; *p* = 0.294). The most striking effect we observed was that the Yazoo isolate had the lowest establishment load in both mosquito species. However, this seems to be an isolate-specific effect, and contrary to expectation (see Figure 1), there was no overall reduction in establishment load associated with infection by drug-resistant vs. drug-susceptible parasite isolates (Figure 2A).

### 2.2. Vector Efficiency

We found that vector efficiency (i.e., proportion of mf that develop to L3 parasites at EIP) differed by parasite isolate and vector isolate, with a significant interaction between these variables (Figure 2B; N = 353; parasite isolate: χ^2^ = 7.434; *p* = 0.059; vector isolate: χ^2^ = 77.422; *p* < 0.001; parasite isolate × vector isolate: χ^2^ = 10.609; *p* = 0.014). However, again contrary to expectation (see Figure 1), there was no overall difference in vector efficiency between mosquitoes infected with drug-susceptible (i.e., GA-2 and Missouri) vs. drug-resistant (i.e., Metairie and Yazoo) parasite isolates. Vector species plays an important role in vector efficiency, with the Metairie isolate of *Ae. albopictus* having very low vector efficiency compared to the Liverpool Blackeye isolate of *Ae. aegypti* (Figure 2B).

### 2.3. Survival to EIP

We found that parasite isolate and vector species interacted strongly to affect mosquito survival to EIP (Figure 2C; N = 1331; parasite isolate: χ^2^ = 80.145; *p* < 0.001; vector isolate: χ^2^ = 1.120; *p* = 0.290; parasite isolate × vector isolate: χ^2^ = 31.512; *p* < 0.001). It is important to recognize that the term “parasite isolate” in this analysis also included the uninfected controls as one category. Thus, the global effects reported here incorporate both the differences in survival between infected and uninfected mosquitoes, as well as differences in survival between the different parasite isolates. One of the most interesting findings of this analysis was the exceptionally large reduction in survival between infected and uninfected *Ae. aegypti* mosquitoes (Figure 2C). However, in the case of *Ae. albopictus*, the magnitude of difference in survival between infected and uninfected mosquitoes was lower, with no significant difference in the case of GA-2 (Figure 2C). Overall, we found that both mosquito species had higher survival when infected by GA-2 compared to other parasite isolates (Figure 2C). However, contrary to expectation (see Figure 1) there was no overall difference in survival to EIP between mosquitoes infected with drug-susceptible (i.e., GA-2 and Missouri) vs. drug-resistant (i.e., Metairie and Yazoo) parasite isolates.

### 2.4. Vectorial Capacity

We found a higher vectorial capacity for *Ae. aegypti* infected by either GA-2 or Yazoo *D. immitis* (one drug-susceptible and one drug-resistant isolate, respectively) when compared to *Ae. aegypti* infected by Missouri or Metairie isolates. In the case of *Ae. albopictus,* we found that mosquitoes infected with Metairie and Yazoo (drug-resistant) isolates had higher vectorial capacity compared to GA-2 and Missouri (the drug-susceptible) isolates. However, this difference was only significant in the case of GA-2 and Metairie. Overall, our data indicated that the most important factor affecting vectorial capacity seems to be the *D. immitis* isolate, with the Metairie isolate-infected *Ae. albopictus* having very low vector efficiency compared to the Liverpool Blackeye isolate of *Ae. aegypti* (Figure 2B).

The structural equation model (SEM) revealed distinct paths affecting vectorial capacity (Figure 3A). Specifically, we found that establishment load positively affected vector efficiency (Figure 3B; β ± SE = 0.623 ± 0.167; F = 13.788; DF = 20.239, *p* = 0.001), and in turn vectorial efficiency negatively affects survival to EIP (Figure 3C; β ± SE = −0.619 ± 0.167; F = 13.660; DF = 20.030, *p* = 0.001). Importantly, we found that vectorial capacity was negatively affected by establishment load (Figure 3D; β ± SE = −0.135 ± 0.057; F = 5.606; DF = 18.091, *p* = 0.029), positively affected by vector efficiency (Figure 3E; β ± SE = 1.257 ± 0.070; F = 322.795; DF = 18.238; *p* < 0.001) and positively affected survival to EIP (Figure 3F; β ± SE = 0.392 ± 0.057; F = 44.472; DF = 19.012; *p* < 0.001). Taking into consideration both direct and indirect effects of the variables affecting vectorial capacity (Figure 3G) we found significant positive total effects of establishment load [β_TOT_ = 0.275; 95% confidence interval (CI): 0.037, 0.467], vector efficiency (β_TOT_ = 0.609; CI: 0.526, 0.700) and survival to EIP (β_TOT_ = 0.304; CI: 0.219, 0.383).

## 3. Discussion

The increase in anthelmintic resistance is a global issue that has critical implications for the current and future efficacy of preventatives and treatments for parasitic diseases of humans and animals [19,25,26,27]. Investigating biological differences between drug-susceptible and drug-resistant parasites and their vectors should provide a better understanding of transmission dynamics and the evolutionary processes involved in the maintenance and spread [11,28] of drug resistance. Such knowledge could facilitate the development of novel treatments and further our understanding of mechanisms of anthelmintic resistance. We found that parasite isolates differed in terms of their initial establishment load, despite the fact that we standardized the mf concentration of each isolate in the blood to 5 mf/µL, and that these isolates had differential effects on mosquito survival (see Figure 2). However, while mosquito mortalities for both *Ae. aegypti* and *Ae. albopictus* were species dependent, there was no discernable effect of drug-resistant or drug-susceptible status. Additionally, both the *D. immitis* isolate and *Ae.* spp. significantly impacted vector efficiency (the proportion of parasites developing to the infective stage at EIP). According to the evolutionary school of thought, we would expect that drug-resistant parasites would have lower fitness than drug-susceptible ones when drug-selective pressures are removed [29]. However, while we found strong parasite isolate and mosquito species interactions on vector efficiency, we found no evidence that the fitness of drug-resistant heartworm is reduced in the mosquito vector.

Drug resistance can have potential life history trade offs that influence parasite transmission. The Yazoo isolate of *D. immitis* exemplified a reduction in initial establishment load (Figure 2A) versus other *D. immitis* isolates. Interestingly, such reductions in initial establishment loads could potentially prove beneficial for reaching infective stages within the mosquito (Figure 2B). Intermediate infection doses maximize vector competence of *D. immitis* [14], which could provide an explanation for how a significant proportion of Yazoo L3s developed in comparison to the other parasite isolates. Another possible explanation to differential L3 development among *D. immitis* isolates is mosquito immunity pathways could play an important role in parasite establishment and development to transmissible stages [30]. However, more experiments are required to clarify whether the initial infective parasite dose of different *D. immitis* isolates has an impact on the development and survival of L3s, and vector parameters.

Our a priori prediction was that drug resistance would consistently reduce establishment load and increase vector efficiency (Figure 1). If the effects of reduced establishment load outweighed the effects of increased vector efficiency, we expected a lower infection intensity for drug-resistant vs. drug-susceptible parasites, and vice versa. In turn, we expected that infection intensity would be negatively correlated with survival to EIP, and thus with vectorial capacity (Figure 1). Thus, based on our theoretical model, we predicted that drug resistance could either increase or decrease vectorial capacity (Figure 1), based on the relative effects of drug resistance on establishment load vs. vector efficiency (Figure 1). However, our observed data revealed idiosyncratic patterns associated with the drug-resistant and drug-susceptible isolates, with apparently no clear association between drug resistance and vectorial capacity (Figure 2D). This finding has critical implications for the spread of filarial drug resistance in natural populations because our data indicate that mosquitoes do not necessarily reduce the transmission efficacy of drug-resistant parasites, unlike in the case of relatively benign parasites such as *Plasmodium* [28,31].

While our data do not support our a priori expectations for the effects of drug resistance on vectorial capacity, they do provide a strong statistical model to better understand the underlying drivers affecting vectorial capacity for filarial parasites. Specifically, our SEM analyses revealed that establishment load had a slight negative direct effect on vectorial capacity (Figure 3G). However, this direct negative effect was overshadowed by a strong positive and indirect effect mediated primarily by the positive effect of establishment load on vector efficiency (Figure 3B). One important caveat of our study design was that mosquitoes were only exposed to a single parasite dose (5 mf/µL). Thus, in our case the positive association between establishment load and vector efficiency likely indicates underlying fitness differences between the parasite isolates. Thus, parasites that had low ability to establish infection initially also had a low likelihood of developing to infective L3 once they established infection. However, previous research has shown that the initial infection dose negatively affects vectorial capacity [14]. Thus, future research will need to test how alterations in parasite dose affects establishment load and vector efficiency. Interestingly, while our SEM model revealed a strong positive effect of vector efficiency on vectorial capacity (Figure 3E), there was significant, albeit weaker, negative effect of vector efficiency on vectorial capacity mediated by the negative effect of vector efficiency on survival to EIP (Figure 3C). Finally, as expected we found a strong positive effect of survival to EIP on vectorial capacity (Figure 3F).

In conclusion, when *D. immitis* infects the vector, the selective pressure of drug treatment is removed for the parasite, and this would be expected to allow drug-susceptible isolates of *D. immitis* to have the comparative advantage in the vector. We were interested in determining whether the development of drug resistance also impacted transmission dynamics by altering vectorial capacity. However, the data did not indicate a consistent pattern for drug-susceptible vs. drug-resistant isolates of *D. immitis* on vectorial capacity (Figure 2) as suggested by our predictive model (Figure 1). Thus, the data presented here indicate that overall patterns of vectorial capacity are potentially more complicated than what a categorization of drug-resistant and drug-susceptible parasites can discriminate, with a stronger effect due to mosquito species, compared to parasite isolates. However, despite the idiosyncratic patterns between the parasite isolates, we were able to elucidate strong causal relationships between the variables affecting vectorial capacity. Specifically, our data demonstrate that in the case of *D. immitis*, transmission risk is primarily driven by a positive indirect effect of establishment load on vectorial capacity, as well as positive direct effects of vector efficiency and survival to EIP on vectorial capacity. Thus, the spread of drug-resistant parasites in the vertebrate population is unlikely to be mitigated by reduced fitness in the mosquito vector.

## 4. Materials and Methods

### 4.1. Simulation Model

To generate quantitative expectations, we simulated the joint effects of establishment risk (i.e., the proportion of microfilariae that initially establish infection in the mosquito) and vector efficiency (i.e., proportion of microfilariae that establish infection that develop to L3 larvae) using the program R 4.0.0 (R Foundation for Statistical computing). We also simulated the probability of survival to the EIP (i.e., 14 days for *D. immitis*) assuming a parasite load-specific mortality using a common logistic function. We calculated vectorial capacity as the product of the three variables simulated. In other words, vectorial capacity is the proportion of microfilariae establishing infection that develop to L3 larvae in mosquitoes surviving to EIP [14]. To simulate the effects of drug resistance on vectorial capacity, we assumed that drug resistance would reduce microfilariae fitness in a drug-free environment (i.e., the mosquito). Thus, we simulated the effects of a reduction of 25% and 75% in establishment risk of the drug-resistant vs. drug-susceptible (i.e., reference) isolates. Additionally, we expected lower pathogenicity of the drug-resistant vs. drug-susceptible isolates, and thus expected higher vector efficiency in the latter vs. the former. We thus simulated the effects of a 25% and 75% increase in vector efficiency of the drug-resistant vs. drug-susceptible (i.e., reference) isolates. We plotted the combined effects of establishment risk, vector efficiency and survival to the EIP on vectorial capacity using a two-dimensional heatmap (as implemented in the R package raster). Further, we predicted the specific expectations for the nine possible outcomes based on all the factorial combinations associated with a 0%, 25% and 75% reduction in establishment risk, and a 0%, 25% and 75% increase in vector efficiency of the drug-resistant isolate compared to the drug-susceptible isolate (which was maintained at the reference value for establishment risk and vector efficiency).

### 4.2. Experimental Procedures

One hundred female *Ae. aegypti* and 60 female *Ae. albopictus* mosquitoes were allocated into separate cages per treatment: uninfected, Missouri, GA2, Metairie, and Yazoo. The mosquitoes were blood fed using a two-chamber inverted glass jacketed membrane feeder (see [14] for details). Briefly, adult female mosquitoes were removed with a mouth aspirator from each cage and placed into a new blood feeding cage. The blood feeding cages had their sugar removed the day prior and water removed at least 4 h prior to blood feeding.

To prepare the blood for feeding mosquitoes, we first estimated the microfilaremia in each infected blood sample by taking two 5 uL aliquots from each sample and counting microfilariae under a compound microscope at 10× magnification. After counts were determined, all infected blood was diluted to a uniform count of 5 mf/uL concentration with uninfected blood. We then added 10 uL of 100mM ATP (adenosine-5′-triphosphate disodium salt, MP Biomedicals, Santa Ana, CA, USA) to each 1 mL of each blood sample as a phagostimulant (see [32] for details) prepared for blood feeders. We dispensed 200 uL aliquots of blood (infected or uninfected as per treatment) into the blood feeders and mosquitoes fed for approximately two hours or until repletion. During feeding, the blood was kept at a constant temperature of 40 °C by circulating hot water through the external chamber of each feeder (see [14] for details). *Ae. aegypti* and *Ae. albopictus* were blood fed the same prepared blood within each treatment (i.e., uninfected or infected with the four *D. immitis* isolates).

The mosquitoes were blood fed following previously mentioned protocols; see above. Blood feeders were removed after feeding and replaced with water and sugar cubes. Afterwards, unfed mosquitoes were removed from the cages. On day five, *Ae. albopictus* mosquitoes were moved into new cages per treatment with a moist paper towel lined cup to allow oviposition. These mosquitoes oviposited for three days and then were moved to new cages per treatment, and maintained until the *D. immitis* microfilariae developed into the infective L3 stage (i.e., the extrinsic incubation period; EIP), which takes approximately 14 days for *Ae. aegypti* [17] and 13 days for *Ae. albopictus* [14].

During and/or after the experiment, the following data were collected with respect to three major parameters: establishment load, vector efficacy and survival to EIP. As mentioned above, we had specific expectations for each of the three variables (see Section 4.1). Thus, we used a power analysis to estimate the required sample size of mosquitoes to detect an effect size equivalent to a 20% change in the response variable as estimated from drug-resistant vs. drug-susceptible (reference category) parasites. We carried out all power analyses by generating simulated data using the *sim.glmm* function implemented in the R package glmmmisc [33], fitting the appropriate statistical model for the response variable (see Section 4.4), and calculating power as the proportion of 1000 simulated datasets for which a significant effect was detected. Below we describe the procedures related to each of the three major parameters we focus on for this study.

#### 4.2.1. Establishment Load

Immediately after blood feeding (1–2 h), we collected a subset of mosquitoes, and these mosquitoes were dissected to determine the observed number of microfilariae (mf) in the mid gut and Malpighian tubules, and thus calculate the initial establishment load. All experiments were performed in triplicate. For the establishment load analyses, we targeted a sample size of 15 mosquitoes (5 from each replicate) per parasite isolate and vector. Our power analyses indicated that this target sample size would give us a power of 0.818 to detect an effect size equivalent to a 20% reduction in establishment load in mosquitoes infected with drug-resistant vs. drug-susceptible parasites.

#### 4.2.2. Vector Efficiency

To estimate vector efficiency, mosquitoes surviving past the EIP (14 days) were collected and dissected to quantify parasite load and characterize the stages of the different parasites detected. In the case of *Ae. aegypti,* we dissected all individuals that survived past the EIP and the experiment was terminated at day 17 post-feeding. In the case of *Ae. albopictus* we dissected all individuals that died after EIP, and to ensure sufficient sample size the experiment was terminated at day 21 post-feeding. During dissection, mosquitoes were categorized by the date of death or was killed, species of mosquito, replicate, and parasite treatment. *D. immitis* were categorized by life stage (mf, L1, L2, or L3) and counted for each dissection. Mosquito dissection and identification of parasite stages followed protocols described earlier [14]. All experiments were performed in triplicate. For the vector efficiency analyses, we targeted a sample size of 45 mosquitoes (15 from each replicate) per parasite isolate per vector. Our power analyses indicated that the target sample size (45 mosquitoes per parasite isolate and vector) would give us a power of 0.900 to detect an effect size equivalent to a 20% increase in vector efficiency in mosquitoes infected with drug-resistant vs. drug-susceptible parasites.

#### 4.2.3. Survival to EIP

To calculate the proportion of mosquitoes surviving to the EIP, we recorded mosquito mortality daily after blood feeding. As mentioned previously, mosquito mortality was monitored for 17 and 21 days, in the case of *Ae. aegypti* and *Ae. albopictus*, respectively. At the end of the experimental period, all remaining mosquitoes were killed and considered to be right-censored data. All experiments were performed in triplicate. For the survival analyses, we targeted an initial population size of 135 blood fed female mosquitoes (45 from each replicate) per parasite isolate. Our power analyses indicated that this target sample size would give us a power of 0.820 to detect an effect size equivalent to a 20% increased probability of survival to EIP in mosquitoes infected with drug-resistant vs. drug-susceptible parasites.

### 4.3. Study System

This study focused on evaluating the vectorial capacity of mosquitoes infected with drug-resistant and drug-susceptible isolates of *D. immitis*, a mosquito-transmitted filarial parasite that primarily infects dogs and other canids. The *D. immitis* life cycle starts when the adult female releases microfilaria (mf) into the blood stream of the vertebrate host, usually a dog, which are subsequently taken up by a mosquito during blood feeding [16]. Microfilariae in the blood meal migrate to the Malpighian tubules, develop into first-stage larvae, and molt twice to reach their infective stage (L3) in the Malpighian tubule cells [15]. Once *D. immitis* reach their L3 stage, they migrate from the Malpighian tubules to the proboscis of the mosquito to be available for when the mosquito takes a blood meal, by entering the exit wound. The time *D. immitis* takes to develop to the L3 stage and reach the head of the proboscis is the extrinsic incubation period (EIP), approximately 14 days for *Ae. aegypti* [17] and 13 days for *A. albopictus* [14].

The *D. immitis* isolates that were chosen include two drug-resistant isolates MET.1 + .2 (2014) and YZO.B.1 + .2 (2013) (hereafter Metairie and Yazoo, respectively) and two drug-susceptible isolates MSO.1 (2000) and GAII.1.2t.3.4 (2013) (hereafter Missouri and Georgia-2, respectively). Detailed descriptions of these isolates have been described elsewhere [24]. The drug-resistant and drug-susceptible isolates of *D. immitis* were obtained from the Filariasis Research Reagent Resource Center and Kaplan laboratory at the University of Georgia.

The vectors used in this investigation were the yellow fever mosquito (*Ae. aegypti*) Liverpool Blackeye strain, and the Asian tiger mosquito (*Ae. albopictus*). *Ae. aegypti* and *A. albopictus* are both natural vectors of *D. immitis* in North America [13]. *Ae. aegypti*, populations used were established from eggs originally obtained from the Filariasis Research Reagent Resource Center [34], a long laboratory line highly susceptible to filarial parasites [35]. *Ae. albopictus*, wild type population, were obtained from Louisiana, 29°59′2.63″ N and 90°7′0.36″ W, and blood fed until repletion, subsequently placed into an insectary at UGA Savannah River Ecology Laboratory. Establishment of the laboratory line from natural populations followed [14]. All experiments described here were undertaken using F3 generation adult mosquitoes, for *Ae. albopictus*. Mosquitoes were maintained under standard insectary conditions: 27 °C, 80 ± 5% relative humidity, and a 12:12 h light diurnal cycle (see [14] for details).

This vector–parasite combination is an ideal model system to study the effects of the mosquito vector on the spread of drug-resistant parasite isolates for several reasons: (a) it is a natural mosquito vector combination [16]; (b) it is logistically easy to work with due to ease of propagation and high infection susceptibility [17]; (c) the system is associated with strong co-evolutionary dynamics because the parasite likely places a large selective pressure on the mosquito due to the high mortality associated with infection [14,36], and in turn the mosquito places a high selection pressure on the parasite, since parasites cannot be transmitted by dead mosquitoes [37]; (d) most importantly, there has been a well-documented and recent rise in drug resistance in *D. immitis* against macrocyclic lactones anthelmintics in the US [19], which has raised serious concerns about the future efficacy of these chemotherapeutic agents [20,21,22]. Drug resistance in *D. immitis* has been suspected for more than 15 years [38] and has been confirmed as a problem for over a decade [19,39,40]. The increasing incidence of resistance is a critical concern because the control of *D. immitis* depends almost exclusively on prophylactic treatment with macrocyclic lactone anthelmintic drugs [22].

### 4.4. Data Analyses

All statistical analyses were carried out using R 4.0.0. Analyses were primarily carried out using generalized linear mixed effects regression (GLMER) as implemented in the R package lme4 [41]. Regression results were modeled using least square means as implemented in the R package emmeans [42].

#### 4.4.1. Establishment Load

Establishment load was defined as the initial number of mf per mosquito immediately after infection. We used a GLMER with a negative binomial error distribution (and log link) to model the initial mf dose as a linear effect of parasite isolate (Metairie, Yazoo, Missouri, GA2), as implemented in the R package lme4 [41]. We included replicate and status (i.e., died or censored) as random factors. 

#### 4.4.2. Vector Efficiency

Vector efficiency was defined as the proportion of developed L3 to the total number of parasites per mosquito surviving the extrinsic incubation period. We used a GLMER with a negative binomial error distribution (and log link) to model the total number of L3 parasites as a linear effect of parasite isolate (Metairie, Yazoo, Missouri, GA2). To control for the effects of establishment load, we included the initial mf dose as a model offset term. All models included replicate and status (i.e., died or censored) as random factors.

#### 4.4.3. Survival to EIP

The probability of a mosquito surviving to EIP was modeled using a GLMER with a binomial error distribution. The status of the individual mosquito (i.e., dead or alive) was used as the dependent variable, and parasite isolate (Metairie, Yazoo, Missouri, GA2) as the independent variable. All models included replicate and status (i.e., died or censored) as random factors.

#### 4.4.4. Vectorial Capacity

We estimated overall vectorial capacity as the joint probability of surviving to EIP and the risk of having L3 in mosquitoes surviving to EIP, following the approach described earlier [14]. Briefly, to calculate overall vectorial capacity we used a zero-inflated negative binomial regression approach and multiplied the estimated vector efficiency (which also controlled for effects of establishment load) with the probability of survival to EIP (see models above). We estimated the standard errors of this measure of vectorial capacity using parametric bootstrap (as implemented in the R package lme4).

Finally, to obtain a better understanding of the causal relationships among the three major variables affecting vectorial capacity (i.e., establishment load, vector efficiency and survival to EIP), we constructed a structural equation model (SEM) ([43]) as implemented in the R package piecewisesem ([44]). The initial model included three equations: (i) vector efficiency~establishment load; (ii) survival to EIP~establishment load + vector efficiency; (iii) vectorial capacity~establishment load + vector efficiency + survival to EIP. We considered replicate a random effect in all models. All the SEM equations were modeled using linear mixed effects models (as implemented in lme4) after z-transformed all continuous variables to obtain standardized model coefficients. To test for potential non-linear effects between the variables we also included quadratic terms of all independent variables in the equations detailed above. We obtained the final SEM by sequentially dropping variables if dropping the variable from a specific model reduced the overall Akaike Information Criterion (AIC) of the SEM model ([45]), and if the removed equation was not considered to be a significant missing path ([44]). Final model acceptance was based on the Fisher’s C statistic, with a model being accepted if the associated *p*-value > 0.05 ([43]). We also calculated the direct, indirect and total effects for each response variables in our final SEM model using the R package semeff ([46]). The final path diagram was plotted using the R package diagram ([47]), and the strength of specific paths was assessed visually using partial residual plots using the R package visreg ([48]).

## Figures and Tables

**Figure 1 pathogens-10-00002-f001:**
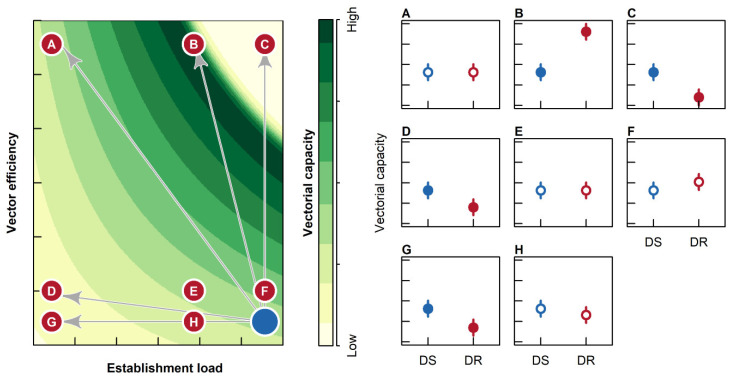
Predicted differences in mosquito vectorial capacity when infected with drug-susceptible (blue symbols) vs. drug-resistant (red symbols) isolates of *D. immitis.* The heat map on the left shows the interacting effects of establishment load (i.e., initial load of establishing parasites) and vector efficiency (proportion of establishing parasites that develop to L3) on vectorial capacity. Vectorial capacity was calculated as the combined product of establishment load, vectorial efficiency and probability of survival to the extrinsic incubation period (EIP). The latter was calculated using a logistic function dependent on predicted L3 parasite load at EIP. We assume that drug resistance will lead to a reduction in establishment load (due to reduced parasite fitness) and increased vector efficiency (due to reduced pathogenicity). These parameters interact to produce a complex set of expectations, with either significant (filled symbols) or non-significant (non-filled symbols) differences between vectorial capacity between drug-susceptible (blue symbols) or drug-resistant (red symbols) parasite isolates (**A**–**H**; see text for details).

**Figure 2 pathogens-10-00002-f002:**
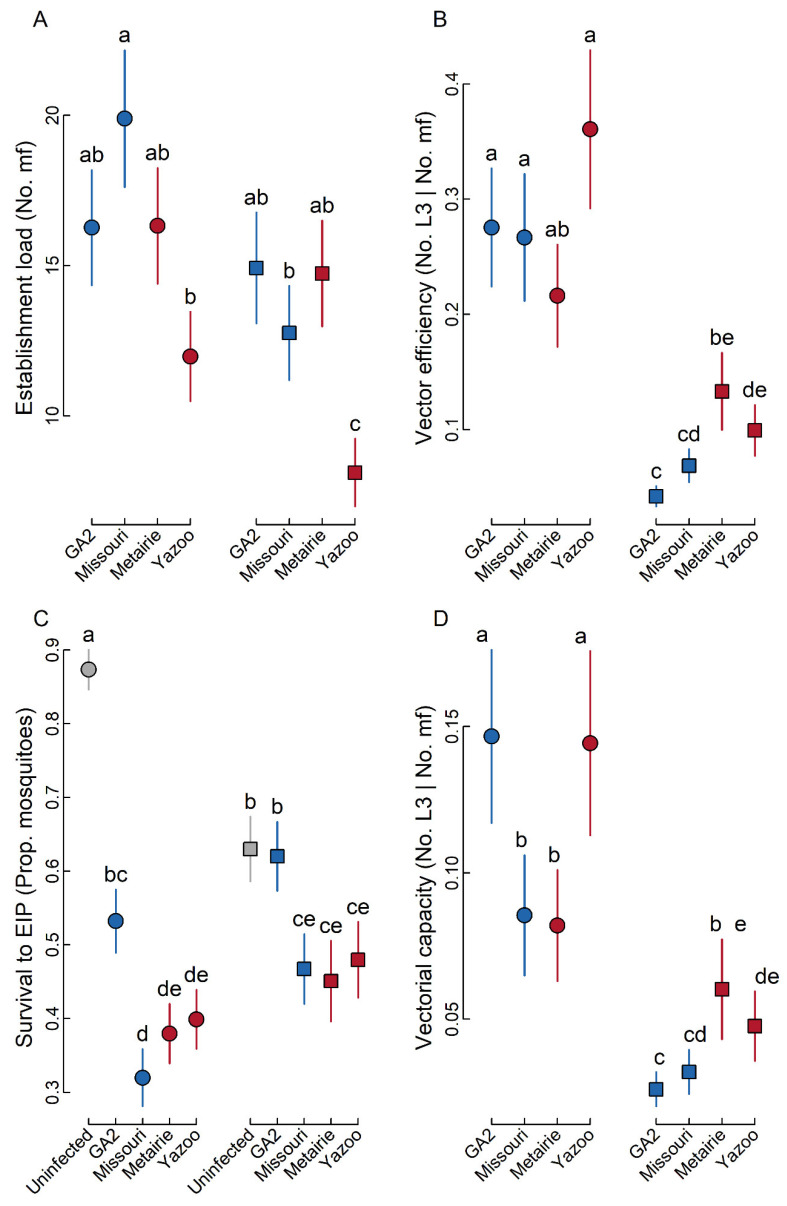
Effects of parasite isolate on parasite transmission risk in *Ae. aegypti* Liverpool Blackeye (**A**–**D**, left side of graphics) and *Ae. albopictus* (**A**–**D**, right side of graphics). Parasite isolates include two drug-susceptible (GA-2 and Missouri; blue symbols) and drug-resistant (Metairie and Yazoo; red symbols) isolates. Separate graphs are given for: (**A**) establishment load measured as the number of viable microfilaria (mf) counted immediately after infection; (**B**) vector efficiency measured as the number of infective L3 parasites developing per mf in mosquitoes surviving to the extrinsic incubation period (EIP); (**C**) survival to the EIP measured as the proportion of surviving mosquitoes; (**D**) vectorial capacity measured as the number of infective L3 parasites developing per mf, accounting for the probability of survival to EIP. Least square means standard error are error bars and least square mean values from the GLMER models show significance when they do not share a letter.

**Figure 3 pathogens-10-00002-f003:**
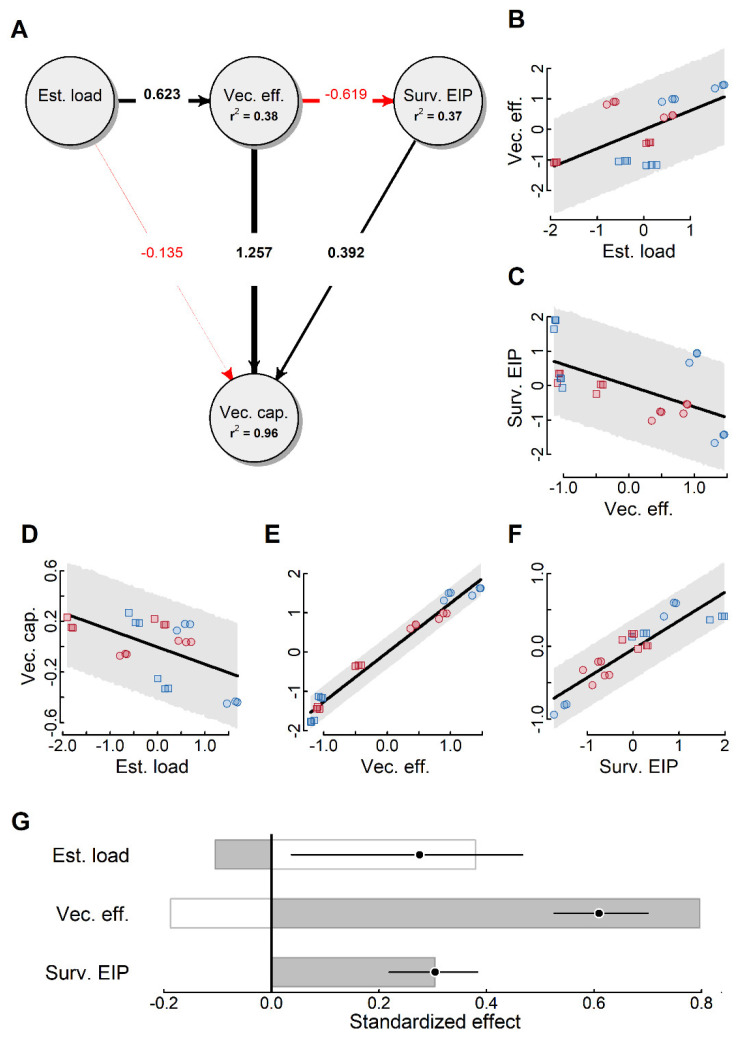
Causal pathways driving *D. immitis* vectorial capacity in *Ae.* mosquitoes. (**A**) The final structural equation model for vectorial capacity (Vect. Cap.) showing the direct and indirect effects of establishment load (Est. load), vector efficiency (Vect. Eff.) and survival to EIP (Surv. EIP). In the path diagram, circles indicate variables (with model r2 values, if applicable). Arrows indicate significant positive (black lines) or negative (red lines) relationships, with standardized coefficients indicated in bold. (**B**–**F**) Partial residual plots of the predictor (x axis) and response (y axis) variables associated with all significant SEM paths. Individual points represent values obtained for each experimental replicate for *Ae. aegypti* (circles) and *Ae. albopictus* (squares) mosquitoes infected by either parasite isolates that were drug susceptible (GA-2 and Missouri; blue symbols) or drug resistant (Metairie and Yazoo; red symbols) with respective 95% confidence intervals (CIs; gray bands). (**G**) Indirect (white bars), direct (gray bars) and total (circles) effects of establishment load, vector efficiency and survival to EIP on vectorial capacity. Error bars are 95% CIs for the estimated total effects.

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
