# Peer review of "Drug Resistance in Filarial Parasites Does Not Affect Mosquito Vectorial Capacity"

_pathogens, 2020, doi:10.3390/pathogens10010002_

Round 1

Reviewer 1 Report

This manuscript and the experiment described are very exciting and interesting. Such information are really missing today and could provide important leads in the management of heartworm resistance. I congratulate the authors for the high quality of their work.

I wonder if the modelization performed could be enhanced by more extensive dataset. Only 15 mosquitoes were dissected per parasite isolate and vector. This represent already a very important workload (could be managed in sequences). Was this "small sample effect" (ie<30) tested from a statistical standpoint in order to check about the variability impact ? If not, this should be discussed.

The impact of the mf concentration in the blood offered to the mosquito is expected to strongly affect the parameters "Establishment load", "Vector efficiency" and "Survival to EIP". The mf blood concentration in the reservoir animals is a parameter known to vary significantly between animals and over time. In the present study, the researchers made the choice to standardize this parameter. This would be interesting to investigate in further research and, at least, to discuss.

Lines 70-76. The police size should be harmonised.

Reviewer 2 Report

In their article ‘Drug resistance in filarial parasites does not affect mosquito vectorial capacity’ the authors describe experimental infection assays of Aedes aegypti and Aedes albopictus mosquitoes with drug-resistant and drug-susceptible Dirofilaria immitis strains, and present models of infection parameters for -resistant and -susceptible parasites lines. In contrast to their hypothesis that drug-resistant parasites would display decreased fitness during the mosquito stages of infection, the authors instead describe stronger parasite strain-specific effects on establishment load, vector efficiency, host survival and vectorial capacity. The study is interesting and represents progress in an important research area, as drug resistance amongst parasite populations is an important issue for vector-borne diseases. However, there are multiple structural issues which make the manuscript difficult to read in places. The manuscript would also be greatly improved if there was more in-depth discussion of the results; comparing the results of the model with those of the experimental infection assays, and if there was additional consideration of the reasons why the presented hypothesis was not supported. Please see my specific comments below.

Major comments:

  1. The introduction is too short, lacks biological information on mosquito-Dirofilaria interactions, and is overly theoretical for a manuscript that is presenting experimental infection data as well as modeling data.

  • It needs to include more information on the course and timeline of infection in mosquitoes (development, EIP, tissue specificity, etc) and the mosquito immune response against the parasite, as these are key drivers of Dirofilaria transmission.
  • There should be some discussion of anti-parasitic drugs, how they act, mechanisms of resistance in parasites, particularly if those details are known for the two resistant lines used in the study. Given that the main hypothesis of the study was not supported, any such differences might explain some of the results.
  • The section discussing the results of the model should be moved from this section to the results section.

  1. I found the discussion section to be rather unsatisfying and lacking analytical depth.

  • There is no discussion of the results of the experimental infection assays in the context of the predictions of the model.
  • No discussion of the implications of the results of the experimental infection assays in the context of disease transmission and drug resistance in mosquito-parasite interactions in nature.
  • Do the drug resistant lines actually have lower fitness? The authors should state explicitly in the discussion what their results suggest about this hypothesis.
  • The Liverpool Aedes aegypti line is a long-term laboratory line, meaning the experiments were not done with an actual F3 generation, but perhaps with the F3 generation in the authors’ lab/insectary. Comparisons of aegypti vs albopictus results in the manuscript should take into account that the differences in laboratory colonization history between the species could have contributed to the differences observed, as this can affect mosquito genetic diversity, development, immunity, microbiome, and fitness, all of which are known to contribute to vectorial capacity. If the albopictus colony had a longer laboratory history it may have behaved more similarly to the aegypti colony.

  1. Given the unexpected results of the experimental infection assays, the authors should seek to understand why their hypothesis was not supported and why they observed strain-specific and mosquito-specific differences in the parasite infection parameters that they measured. Given that they four parasite strains that were used are all reference lines obtained from a stock centre, it may be possible to perform some comparative genomics, or at least to speculate about genetic differences between strains. The authors could also investigate the mosquito side of the equation and perform some basic immunity assays on a limited number of interesting mosquito species-parasite strain combinations. This could include quantification of key immune pathway genes or levels of reactive oxygen species.

  1. No data on the parasite-free control groups are included in survival assay results.

  1. The text is unclear on the number of experimental replicates that were performed for all assays. Line 300 states that “All experiments were performed in triplicate.”, but this is placed in the section on survival experiments and it is unclear if it applies only to that sectino. This issue should be clarified in the text.

Minor comments

  1. Line 26 – should this read species and not strain?

  1. Line 36 – malaria cases have declined and then stagnated over the past decade.

  1. Line 151-152 – “We found an increase of vectorial capacity for Ae. aegypti infected by either GA-2 or Yazoo D. immitis”– The term increase is imprecise as you are not able to compare against a control group.

  1. Lines 158-160 – “This section may be divided by subheadings. It should provide a concise and precise description of the experimental results, their interpretation as well as the experimental conclusions that can be drawn.” – formatting text to be removed.

  1. The paragraph starting line 219 on the rationale of mosquito/parasite choices should be moved to the introduction or discussion sections.

  1. Fig. 2 legend – From the text in the legend, it is unclear which graphs denote Aedes aegypti and which denote Aedes albopictus. It is also unclear if group means or medians are presented. Finally, the text should state whether the data in the figure represent the average of replicates or data from a representative replicate, assuming all assays were replicated (see comment 5).

  1. The text describes the survival experiments as being ‘to’ and ‘through’ the EIP. These have entirely different meanings and the text should be altered accordingly to reflect which assay was actually performed.

Reviewer 3 Report

This manuscript attempts to measure the cost of increased drug resistance in the dog heartworm Dirofilaria immitis regarding 1) establishment load of filaria in the vector mosquito, 2) proportion of microfilaria that survive to L3 in the mosquito, and 3) survival of mosquito through external incubation period. Variance between mosquito strain and pathogen strain turned out to be the only significant variables.

This experiment is generally well-designed and well-written. The subject matter is certainly of great interest, both from the point of view of veterinary medicine and evolutionary biology.

My primary concern is the lack of depth in the Discussion in interpreting the results. How could the variance in the sampling method have influenced the results? How could differences in mosquito strain influenced the results? How are these results interpreted compared to similar experiments done elsewhere, perhaps on other systems?

Specific comments can be found below.

INTRODUCTION

  1. Is the first time anybody has looked at trade-offs of drug resistance in a mosquito/Dirofilaria system? The Introduction cites a number of Plasmodium references, but there no mention of Dirofilaria. Any other work should be cited.
  2. It would be good to see a bit more background given on Dirofilaria immitis. Even a sentence or two citing 1-3 general references, just to introduce the parasite to non-experts.

RESULTS

3. Lines 158-160 – it looks like text from the authors guidelines hasn’t been deleted from the manuscript.

DISCUSSION

4. How much could the variance of the initial microfilaria sampling method have contributed to the variance in the results?

5. Would there have been much genetic variation in any of the mosquito strains, particularly with any that had been recently colonised from wild mosquitoes?

MATERIALS AND METHODS

6. 209-210. I could not access reference #23 (McCall 1981), but reference #24 (Bowman et al 2009) makes no reference to an Aedes albopictus colony. Why were these cited? If they were cited just to support that Ae. aegypti and Ae. albopictus are vectors of D. immitis, that should go in Introduction.

7. 211-212. “FR3” should be spelled out for those not familiar with filiarial research

8. 211-212. Reference #25 doesn’t give much of a history of the Aedes aegypti strain. Where was it colonised from? How many years/generations has it been in colony?

9. 212-215. To be clear, the same strains used from reference #11 (Dharmarajan et al 2019) were used in this study? If so, how long had they been in colony when they were used in this study?

10. 215-216 “All experiments were done with F3 generation adults”—does that mean F3 from when they were taken out of the colony, or F3 from when they were colonised from the wild?

11. Lines 219-231. This section is more of a justification of methods as opposed to pure description of methods. The authors and/or editor may disagree, but you might consider moving this to the Introduction section, as part of setting up a description of the goals of the study. Alternatively, it might be in the Discussion.

12. Lines 261-263. Is the method for estimating microfilaria density well established? If so, please provide a reference. I see this estimation method—specifically, the variance of estimations using this method--as critical to the results of the paper. Random variation in the initial microfilaria densities of the mosquitoes may explain the results.

13. Line 272. What are “Mosquito Blood Feeding” and “Blood Treatment” protocols? Since those words are italicised, I assume they are titles, but I cannot find them in the references section. Please cite more clearly.

Round 2

Reviewer 2 Report

The majority of my comments have been addressed to my satisfaction. The others represent differences of opinion rather than serious issues, so I am comfortable with the manuscript being accepted for publication.